# The Influence of Different Guided Bone Regeneration Procedures on the Contour of Bone Graft after Wound Closure: A Retrospective Cohort Study

**DOI:** 10.3390/ma14030583

**Published:** 2021-01-27

**Authors:** Maoxia Wang, Xiaoqing Zhang, Yazhen Li, Anchun Mo

**Affiliations:** 1State Key Laboratory of Oral Diseases & National Clinical Research Center for Oral Diseases & Department of Oral Implantology, West China Hospital of Stomatology, Sichuan University, Chengdu 610041, China; wangmaoxia@stu.scu.edu.cn (M.W.); 2017224035151@stu.scu.edu.cn (X.Z.); 2State Key Laboratory of Oral Diseases & National Clinical Research Center for Oral Diseases & Department of Orthodontics, West China Hospital of Stomatology, Sichuan University, Chengdu 610041, China; 2019324035095@stu.scu.edu.cn

**Keywords:** guided bone regeneration, wound closure, computer-aided surgery, lateral ridge augmentation, bone defect

## Abstract

The aim of this study was to evaluate the impact of different guided bone regeneration (GBR) procedures on bone graft contour after wound closure in lateral ridge augmentation. A total of 48 patients with 63 augmented sites were included in this study. Participants were divided into 4 groups (*n* = 12 in each group) based on different surgical procedures: group 1: particulate bone substitute + collagen membrane; group 2: particulate bone substitute + collagen membrane + healing cap, group 3: particulate bone substitute + injectable platelet-rich fibrin (i-PRF) + collagen membrane; group 4: particulate bone substitute + i-PRF + surgical template + collagen membrane. After wound closure, the thickness of labial graft was measured at 0–5 mm apical to the implant shoulder (T0–T5). At T0–T2, the thickness of labial graft in group 4 was significantly higher than the other three groups (*p* < 0.05). And group 4 showed significantly more labial graft thickness than group 1 and group 2 at T3–T5 (*p* < 0.05). Within the limitations of this study, the use of i-PRF in combination with the surgical template in GBR may contribute to achieving an appropriate bone graft contour after wound closure.

## 1. Introduction

After tooth extraction, alveolar ridge resorption is initiated and leads to inadequate bone dimensions for implant placement [1]. In this situation, bone augmentation is necessary to reconstruct the ideal alveolar ridge contour and obtain adequate bone for implant placement. Various techniques have been proposed for bone augmentation, such as autogenous bone block grafting, distraction osteogenesis, ridge expansion and guided bone regeneration (GBR) [2,3]. GBR with particulate graft materials and resorbable collagen membranes (CM) has become a routine procedure for the repair of defects and renders long-term clinical success [4,5,6,7,8,9].

However, A major drawback of GBR with particulate bone substitutes and CM is their poor graft stability [4]. The soft tissue pressure following wound closure can lead to an apical displacement of bone grafts, resulting in a reduction of horizontal thickness in the coronal portion of the augmented site, while the contour of bone graft after wound closure determines the initial space for new bone formation [10,11]. And the compressive forces of soft tissue generated during the healing period may result in a further collapse of the grafts, a sufficient over-augmentation of the bone defect after wound closure is therefore needed to compensate for the displacement and resorption of bone grafts [4]. Moreover, based on the concept of “prosthetic guided regeneration”, the reconstruction of bone and soft tissue contours should be carried out under the guidance of preoperatively designed prosthetic restorations and implants [12]. Thus, obtaining adequate bone graft around the implant after wound closure in a precise and reproducible manner is still challenging in lateral ridge augmentation.

Several strategies have been applied to achieve a better contour of bone graft, such as using fixation pins or healing cap to stabilize membranes, or mixing particulate bone substitutes with injectable platelet-rich fibrin (i-PRF) to form “sticky bone”, which can enhance the volume stability and operability of grafts [4,13]. An in vitro study has compared graft stability of different GBR techniques in the treatment of three-wall horizontal bone defects, it was found that applying pins for the fixation of collagen membrane can enhance the stability of particulate graft materials and reduce the apical displacement of bone grafts [14]. Mertens et al. reported similar results in the treatment of one-wall horizontal bone defects [15]. However, as these in vitro studies only partially simulated clinical situations, there is very limited clinical evidence about the effect of different GBR procedures on the contour of graft after wound closure. Furthermore, applying digital technology to GBR procedures may help to improve the predictability of bone augmentation. Several case reports have reported the successful application of 3D-printing individualized titanium mesh and individualized allogeneic bone blocks in the treatment of severe bone defects [16,17,18]. However, there is still a lack of effective and convenient digital methods in the treatment of small and moderate bone defects.

This study aimed to analyze whether the use of (1) particulate bone substitute +CM exhibit different results from (2) particulate bone substitute + CM + healing caps, (3) sticky bone + CM + pins and (4) sticky bone + surgical template + CM + pins with respect to the contour of bone graft after wound closure. The null hypothesis was that there is no difference between these GBR procedures.

## 2. Materials and Methods

### 2.1. Study Population

This study was designed as a retrospective cohort study. All patients included in this study were treated at the Department of implant dentistry, West China Hospital of Stomatology, Sichuan University, China from January 2016 to December 2019. The study was conducted following the Helsinki declaration and approved by the ethical committee of West China Hospital of Stomatology, Sichuan University (WCHSIRB-D-2019-068).

#### 2.1.1. Inclusion Criteria

Male or female patients aged 18 to 65 years (including 18 and 65 years).Presence of a three-wall or two-wall horizontal bone defect in the anterior region.Received one of the following GBR procedures: particulate bone substitute +CM (Group 1), particulate bone substitute + CM + healing caps (Group 2), sticky bone + CM + pins (Group 3), or sticky bone + CM + pins + surgical template (Group 4).Bone augmentation was applied > 3 months after tooth extraction.Good general health and absence of periodontal diseases.Patients were willing to participate in this study and signed the informed consent form.

#### 2.1.2. Exclusion Criteria

Uncontrolled systemic diseases.Heavy smokers (>20 cigarettes per day).Females in pregnancy or lactation.

### 2.2. Surgical Procedures

An experienced surgeon (MA) performed all the surgical procedures. All patients received antibiotic prophylaxis (2 g amoxicillin) one hour before surgery. Patients were instructed to rinse with 0.5% povidone–iodine 1 min three times to disinfect the surgical site. After local anesthesia (4% solution of articaine), A mid-crestal and two vertical releasing incisions were performed, and a full thickness mucoperiosteal flap was elevated. The implant sites were prepared according to the manufacturer’s recommendations. Implants (NobelActvie, Nobel Biocare, Göteborg, Sweden; Bone Level Titanium SLA, Institut Straumann AG, Basel, Switzerland; Tapered Screw-Vent MTX, Zimmer Dental Inc., Carlsbald, CA, USA; Axiom REG, Anthogyr, Sallanches, France) of proper size were placed in a prosthetically ideal position. A staged approach of implant placement was performed at the sites without enough primary stability. One of the following GBR procedures was performed at the defect areas.

Group 1: After cortical perforation and implant placement under the guide of a digital implant template, a cover screw was connected to the implant. The defect was filled with particulate bone substitutes (Bio-Oss, Geistlich Pharma AG, Wolhusen, Switzerland; Bio-Gene, Beijing Datsing Bio-tech Co., Beijing, China) mixed with physiological water to achieve 1 mm of over-augmentation for the buccal contour of the alveolar ridge. An appropriately sized collagen membrane (Bio-Gide, Geistlich Pharma AG, Wolhusen, Switzerland) was selected and trimmed to adapt to the defect size, and it was used to cover the bone grafts with an overlap of 2 mm (Figure 1).

Group 2: The treatment was similar to group 1. However, a wide healing cap was connected to the implant to stabilize the bone substitutes and membrane (Figure 2).

Group 3: Cortical perforation was performed following flap elevation. For the patients with simultaneous implant placement, a cover screw was connected to the implant after implant placement. Blood was taken from the elbow vein and centrifuged at 700 rpm for 3 min to obtain i-PRF. Then particulate bone substitutes were mixed with i-PRF to form sticky bone [13]. The sticky bone was placed into the defect to achieve 1 mm of over-augmentation and covered with a collagen membrane. In addition, titanium pins were used to fixate the membrane (Figure 3).

Group 4: The treatment was similar to group 3. However, a two-piece tooth-supported surgical template was fabricated through 3D printing technology (ProJet MJP 2500Plus, 3D Systems, Inc., Rock Hill, SC, USA) before surgery base on the digital simulation of bone graft contour (Mimics 20.0, Materialise, Leuven, Belgium). And the mixture of i-PRF and particulate bone substitutes was placed into the defect under the guidance of the template to form customized sticky bone. Next, the customized sticky bone was cover with collagen membrane and fixed with pins (Figure 4).

A periosteal release incision in the apical region of the labial flap was performed to achieve a tension-free primary closure. The flaps were sutured with horizontal mattress sutures and single interrupted sutures. Cone beam computed tomographic (CBCT) scans were carried out immediately after wound closure.

The patients were instructed to rinse their mouth with 0.12% chlorhexidine three times 1 day and prescribed post-operative antibiotics for 5 days (3 × 750 mg amoxicillin/day). The sutures were removed 14 days post-surgery.

### 2.3. CBCT Analysis

All the patients underwent CBCT scanning before surgery and immediately after wound closure using the same projection condition (3DAccuitomo 170, J. Morita Mfg. Corp., Kyoto, Japan). The technique parameters were as follows: acceleration voltage, 90 kV; beam currency, 5 mA; acquisition time: 17.5 s; FOV diameter, 140 mm; FOV height, 100 mm; and voxel size, 0.25 mm.

Simplant software (Simplant Pro 17.01, Dentsply Sirona, Philadelphia, PA, USA) was used to evaluate the digital imaging and communications in medicine (DICOM) datasets. According to the concept of “prosthetic guided regeneration”, a 3Shape TRIOS^®^ intraoral scanner (3Shape, Copenhagen K, Denmark) was used to obtain maxillary and mandibular digital impressions of each patient before surgery. Then a diagnostic wax-up was made based on the intraoral scan data. Subsequently, the DICOM files of the preoperative CBCT radiographs and the Standard Tessellation Language (STL) files of the diagnostic wax-up were superimposed in Simplant software. 3.5-mm-diameter and 10-mm-length virtual implants were placed in a prosthetically ideal position under the guidance of the diagnostic wax-up [19,20]. After that, the DICOM files of the postoperative CBCT were converted to STL files and superimposed on the preoperative CBCT. In the bucco-oral cross-sectional image perpendicular to the virtual implant, the distance between the implant and the labial outline of graft, which represented the thickness of the labial bone graft, was measured at the implant shoulder (T0) and at 1, 2, 3, 4 and 5 mm (T1–T5) apical to the implant shoulder (Figure 5). The standard deviation of T0, T1 and T2 (SDC) at each site, which represents the uniformity of the graft contour in the coronal portion, was calculated for each site.

A trained investigator (LY) designed the position of implants and performed all measurements in a blinded manner. And all measurements were performed twice and averaged.

### 2.4. Sample Size Calculation

Sample size calculation was performed using a sample size software (PASS 15, NCSS, LLC. Kaysville, UT, USA). In this study, we choose T0 as the primary outcome. To achieve 80% power to detect differences among the means versus the alternative of equal means using an F test with a 0.05 significance level, 12 patients per group (total of 48 patients) were required. The size of the variation in the means was represented by the effect size f = σm/σ, which was 0.5.

### 2.5. Statistical Analysis

Statistical analyses were conducted using SPSS software version 20.0 (IBM company, Armonk, NY, USA). All the parameters were summarized by descriptive statistics. The data were reported by means, standard deviations (SD) and 95% confidence intervals (95% CI). Chi-square tests were performed for categorical variables. The normality of continuous variables was checked using Shapiro-Wilk test. The assumption of homogeneity of variance was examined using Levene’s test. All the continuous variables fulfilled normal distribution and homogeneity of variance. One-way analysis of variance (ANOVA) was applied to determine age differences between the groups. Multi-factor ANOVA was used to evaluate the effect of GBR procedures, location, jaw, defect type, implant type and bone substitutes, and Student–Newman–Keuls test was used for a comparison between two groups. The statistical test level was set as 0.05.

After a 4-week interval, 10 randomly selected sites were remeasured to determine intra-observer reliability. Intra-observer reliability was performed by evaluating the intra-class correlation coefficient (ICC) for T0–T5 that resulted in good agreement (ICC ranged from 0.969 to 0.995).

## 3. Results

A total of 48 patients with 63 augmented sites were included in this study. The characteristics of the patients and augmented sites are summarized in Table 1. No statistically significant differences existed in gender, age, jaw and defect type. Significant differences were found in location, implant type and bone substitutes (*p* < 0.05).

The results of Multi-factor ANOVA are displayed in Table 2. Multi-factor ANOVA revealed that location, jaw, defect type, implant type and bone substitutes didn’t lead to many confounding effects to the contour of bone graft after wound closure, while the significant effect of GBR procedures was found (*p* < 0.05). The results of the thickness of labial bone graft at the 6 different levels (T0–T5) for the four GBR procedures are presented in Table 3 and Figure 6. At T0–T2, group 4 (mean ± SD: 4.29 ± 0.92 mm at T0, 4.54 ± 0.88 mm at T1, 4.68 ± 0.79 mm at T2) showed significantly more labial graft thickness than group 1, 2 and 3. At T3–T5, group 4 (mean ± SD: 4.85 ± 0.82 mm at T3, 4.76 ± 0.96 mm at T4, 4.72 ± 0.98 at T5) and group 3 had a considerably better outcome than group 1 and 2. And there were no significant differences between group 1 and group 2 at all levels.

In terms of the uniformity of the graft contour in the coronal portion, group 2 (mean ± SD: 0.18 ± 0.09 mm) and group 4 (mean ± SD: 0.20 ± 0.09 mm) showed a better result than group 1 (mean ± SD: 0.31 ± 0.17 mm) and group 3 (mean ± SD: 0.44 ± 0.15 mm), the difference was statistically significant (Figure 7). This indicates that group 2 and group 4 appeared less collapse in the coronal portion of bone grafts. And group 1 showed a better outcome than group 3.

## 4. Discussion

Our results demonstrated that GBR with customized sticky bone can provide a greater thickness of labial bone graft after wound closure, especially in the coronal portion, when compared with the other three GBR procedures. Moreover, GBR with customized sticky bone and GBR with particulate bone substitutes and a wide healing cap exhibited less collapse in the region of the implant shoulder.

Establishing aesthetics and achieving long-term success is challenging in implant therapy. Adequate labial graft thickness is essential to maintain stable crestal bone level and prevent marginal soft tissue recession, and the existence of more than 2mm labial graft thickness has been advocated for long-term stable implant success [21,22]. As the contour of bone grafts after wound closure forms the initial space for new bone regeneration, it’s important to obtain sufficient graft volume in lateral ridge augmentation. A meta-analysis of 15 randomized controlled trials found that the estimated mean (±SD) bone resorption during 6-months healing period was 1.22 ± 0.28 mm for GBR in lateral ridge augmentation [23]. Thus, it can be speculated that an approximately 3.5 mm thickness of labial graft at all levels after wound closure is needed to compensate for the future loss of graft volume in lateral ridge augmentation procedure.

In view of the meaning and role of bone graft contour after wound closure, four different GBR procedures were compared in this in vivo study. In the present study, two parameters, the labial graft thickness (T0–T5) and the standard deviation of T0, T1 and T2 (SDC), were used to assess the clinical outcomes of these four procedures. As SDC represented the uniformity of the graft contour in the coronal portion, it thus indicates the degree of collapse in the coronal portion of the bone graft.

In group 1, the most used GBR technique–GBR with particulate bone substitutes and CM–was performed. However, our study showed that there is an obvious coronal collapse of bone grafts in group 1, the labial graft thickness in the coronal portion (2.69 ± 1.00 mm at T0, 2.97 ± 0.86 mm at T1) is insufficient. Similar results have been reported in other studies. Mir-Mari et al. reported in an in vitro study assessing the influence of wound closure on the volume stability of different GBR procedures that wound closure can induce a significant apical displacement of particulate bone substitute materials in the treatment of three-wall horizontal defect [24]. The postoperative collapse of grafts at the level of the implant shoulder appears to be common in GBR procedures [11,14,15,25,26]]. These results may be explained by the poor mechanical properties of the particulate grafting material and the resorbable collagen membrane [4,27]. The soft tissue pressure following wound closure can cause the collapse of the collagen membrane and particulate bone substitutes. Thus, additional methods were needed to enhance the stability of bone grafts.

In group 2, a wide healing cap was used to support the resorbable collagen membranes [4]. In the present investigation, the degree of the coronal collapse of bone grafts was significantly reduced in group 2 in comparison with group 1. This finding proved that the use of a wide healing cap has a contribution to stabilizing collagen membrane and particulate bone substitute. However, there was no statistically significant difference in labial graft thickness at all levels (T0–T5) between groups 1 and 2. This is potentially caused by the fact that the diameter of the healing cap is usually 1–2 mm larger than the diameter of implants. And the healing cap is difficult to provide enough space for bone grafts. Another possible reason for this is that the operator may fail to achieve an appropriate bone graft contour before wound closure, while all the GBR processes were under freehand operation.

As mentioned above, the mechanical properties of grafting materials and membranes have a big impact on the contour of bone graft after wound closure. While particulate grafting materials exhibited insufficient volume stability, improving the mechanical performance of particulate grafting materials seems to be a better choice. In groups 3 and 4, particulate bone substitutes were mixed with i-PRF to form sticky bone for bone augmentation [13,28]. As graft particles are incorporated in the fibrin matrix from i-PRF, the compression capability of the particulate materials was enhanced. Scarano et al. found that adding autologous platelet liquid to bovine bone granules led to a composite scaffold with increased compressive resistance of 175% in comparison with the mixture of blood and bovine bone particles, and increased compressive resistance of 875% compared with the mixture of physiological water and bovine bone particles [29]. Another advantage of sticky bone is that this soft-type graft block can easily be shaped and adapted to the defect site, while the proper trimming and placement of bone blocks can be challenging to operators [30]. In addition, i-PRF contains plentiful regenerative cells and can provide a sustained release of growth factors, including transforming growth factor-beta1, platelet-derived growth factor and vascular endothelial growth factor, which contribute to wound healing and tissue regeneration [31,32,33,34,35].

However, the present study showed different results between groups 3 and 4. Group 3 exhibited obvious coronal collapse of bone grafts similar to group 1. And there was no statistically significant difference in labial graft thickness at level T0–T2 between groups 1, 2 and 3. In contrast, group 4 showed significantly less coronal collapse of bone grafts in comparison with groups 1 and 3. In addition, labial graft thickness at all levels (T0–T5) was almost more than 3.5 mm in group 4. The use of customized sticky bone for GBR was associated with significantly more labial graft thickness at all levels, as compared to GBR with particulate bone substitutes and GBR with particulate bone substitutes in combination with a wide healing cap. And GBR with customized sticky bone showed significantly more labial graft thickness in the region of the implant shoulder in comparison with GBR with sticky bone.

This intriguing result could be due to the use of the surgical template. Despite the fact that sticky bone can provide better volume stability, the contour of bone graft is also affected by human operation. It is noteworthy that the standard deviations of SD_C_ in group 3 were bigger in comparison with group 4, which indicated that the results of freehand operation were not sufficiently stable, although all the surgeries were performed carefully by the same experienced surgeon. In group 4, the mixture of i-PRF and particulate bone substitutes can be properly shaped under the guidance of a surgical template. The combination of sticky bone and surgical template provided a stable and appropriate contour of bone grafts during the surgery.

With regard to the research methods, some limitations need to be acknowledged. different types of bone substitutes, location and implant were involved in this study. However, the particle size of these two bone substitutes is approximately the same (the particle size of Bio-Oss granules is 0.25–1 mm, and the particle size of Bio-Gene granules is 0.05–1 mm). Thus, the discrepancies between these two types of bone substitutes may have little impact on the study results, as the scope of this research was the contour of bone graft after wound closure. And we have performed Multi-factor ANOVA to evaluate the confounding effects from these factors. Results revealed that location, implant type and bone substitutes did not cause many confounding effects on the contour of bone grafts. In addition, blind methods were applied to the outcome assessor to reduce the risk of bias. The reliability evaluation also demonstrated that all the measurements were highly repeatable in this study, as the ICC values for all these measurements were higher than 0.95. And our results were limited in a two-dimensional manner, while the two-dimensional results may be more effective to provide guidance in clinical practice when compared with three-dimensional information. Moreover, as the presented study focuses on the contour of bone grafts after wound closure, no firm conclusions can be drawn about the long-term stability of bone grafts. Further investigations are required to confirm the amount of new bone formation and the long-term stability of bone grafts under treatment with these GBR procedures.

## 5. Conclusions

Within the limitations of the study, the following conclusions can be drawn:GBR with customized sticky bone showed better results than GBR with particulate bone substitutes, GBR with particulate bone substitutes in combination with a wide healing cap and GBR with sticky bone with respect to the thickness of labial graft immediately after wound closure.The use of customized sticky bone and wide healing cap enhanced volume stability of bone grafts, especially in the coronal portion of bone grafts.The use of surgical templates contributed to an appropriate contour of bone grafts after wound closure.

## Figures and Tables

**Figure 1 materials-14-00583-f001:**
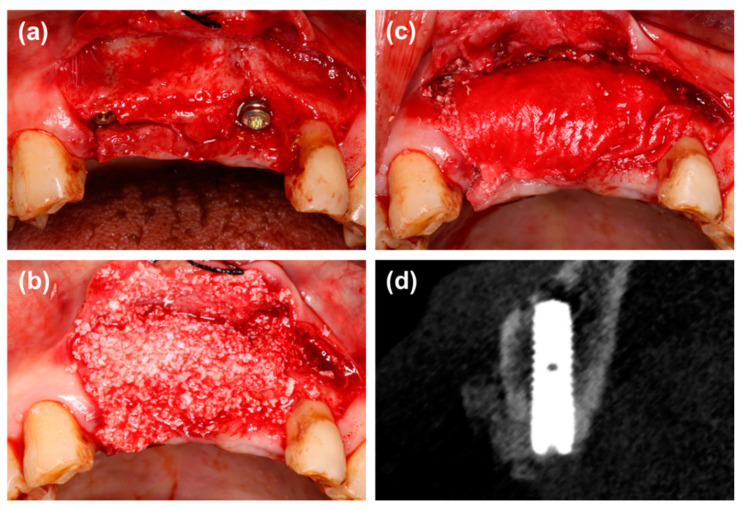
(**a**) The labial defect could be observed. (**b**,**c**) Guided bone regeneration with particulate bone substitutes and collagen membrane. (**d**) Radiographic view of cone beam CT immediately after wound closure.

**Figure 2 materials-14-00583-f002:**
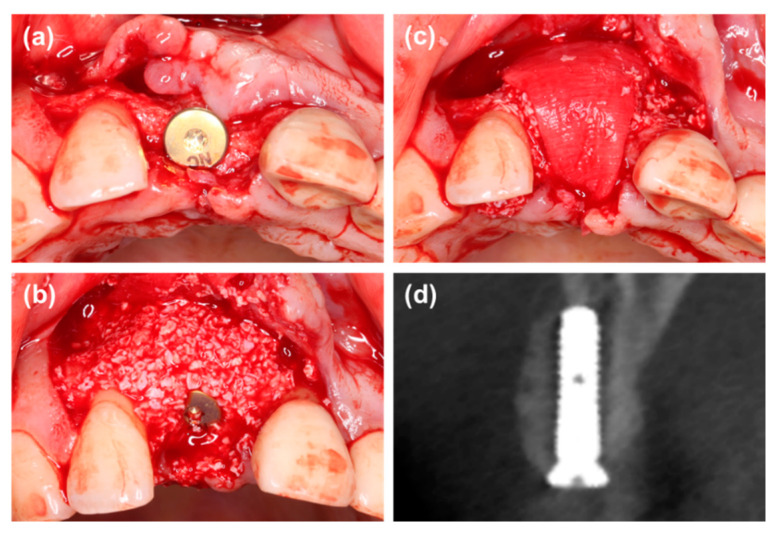
(**a**) The labial defect could be observed, and a wide healing cap was connected to the implant. (**b**,**c**) Guided bone regeneration with particulate bone substitutes and collagen membrane. (**d**) Radiographic view of cone beam CT immediately after wound closure.

**Figure 3 materials-14-00583-f003:**
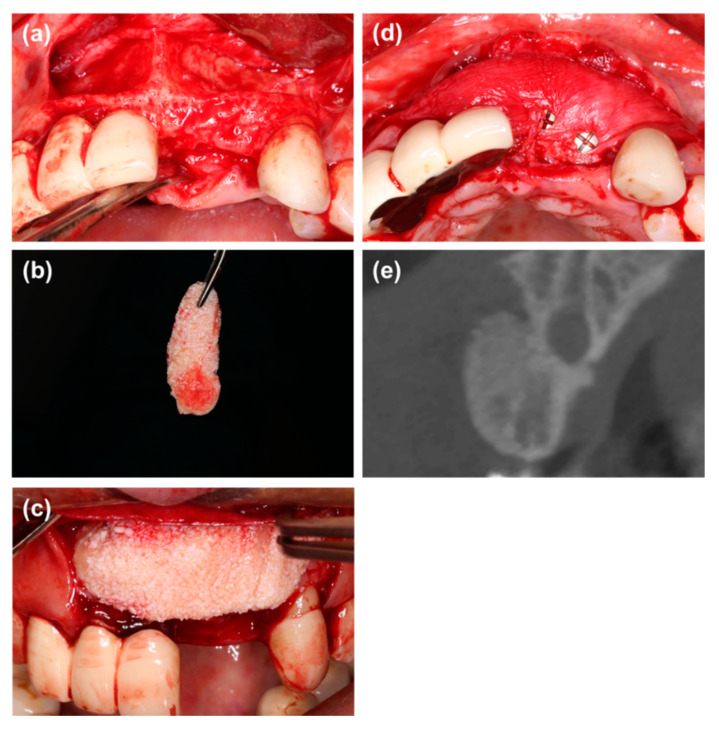
(**a**) The labial defect could be observed. (**b**) Particulate bone substitutes were mixed with i-PRF to form sticky bone. (**c**,**d**) The sticky bone was placed into the bone defect and covered with a collagen membrane, and titanium pins were used to fixate the membrane. (**e**) Radiographic view of cone beam CT immediately after wound closure.

**Figure 4 materials-14-00583-f004:**
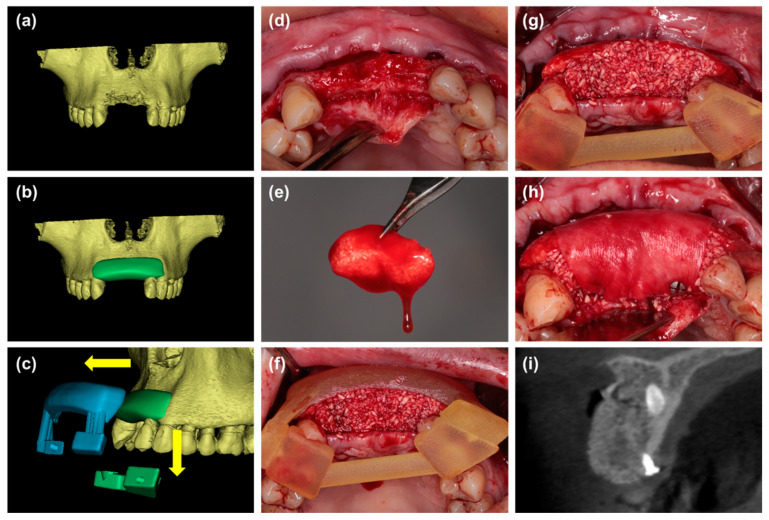
(**a**,**b**) Digital simulation of bone graft contour before surgery. (**c**) A two-piece surgical template, which consists of two parts: the coronal part (green) for retention and the labial part (blue) for shaping the bone grafts, was fabricated based on the digital model. the template can be removed without disrupting the graft material. (**d**) The labial defect could be observed. (**e**) Particulate bone substitutes were mixed with i-PRF. (**f**,**g**) The mixture of i-PRF and particulate bone substitutes was placed into the defect under the guidance of a surgical template to form customized sticky bone. (**h**) The customized sticky bone was covered with a collagen membrane and fixed with pins. (**i**) Radiographic view of cone beam CT immediately after wound closure.

**Figure 5 materials-14-00583-f005:**
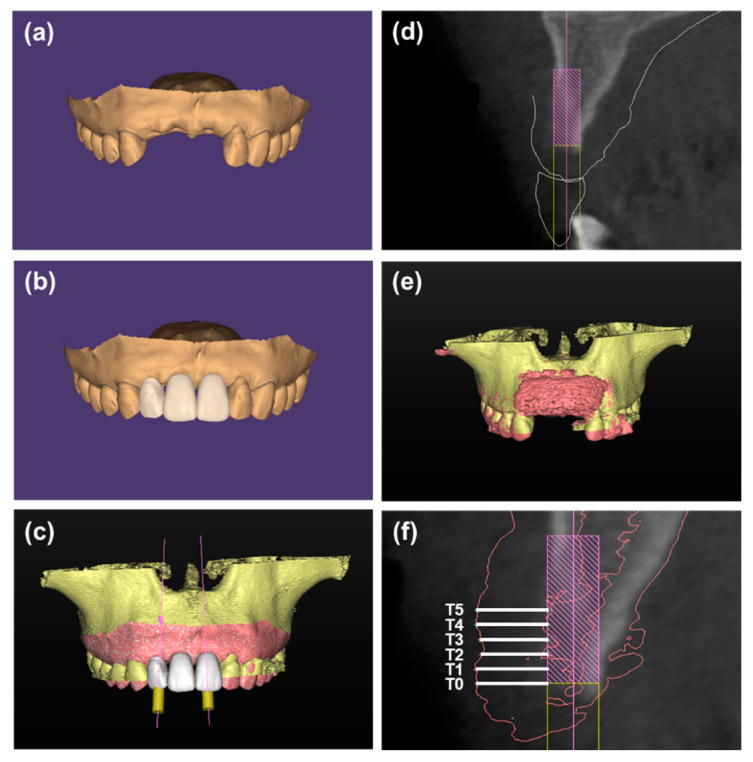
(**a**–**d**) Virtual implants were placed in a prosthetically ideal position under the guidance of the diagnostic wax-up. (**e**) The DICOM files of the postoperative CBCT were converted to STL files (red) and superimposed on the preoperative CBCT (yellow). (**f**) The thickness of labial bone graft was measured from the labial outline of the bone graft (red lines) to the implant (T0–T5).

**Figure 6 materials-14-00583-f006:**
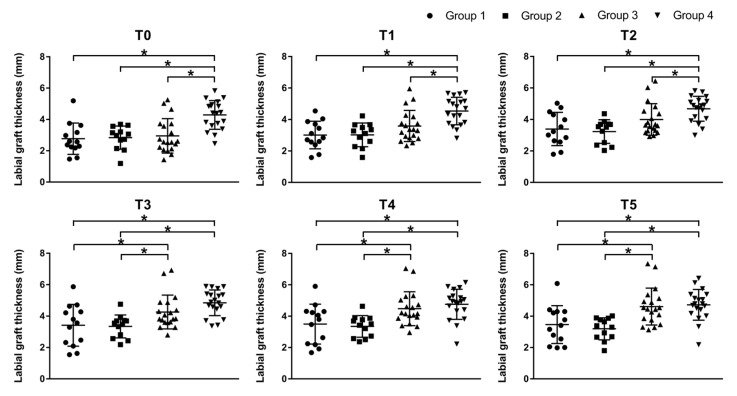
Quantification of labial graft thickness at different apico-coronal levels (T0–T5) immediately after wound closure. Error lines represent +/− standard deviation. * Statistically significant.

**Figure 7 materials-14-00583-f007:**
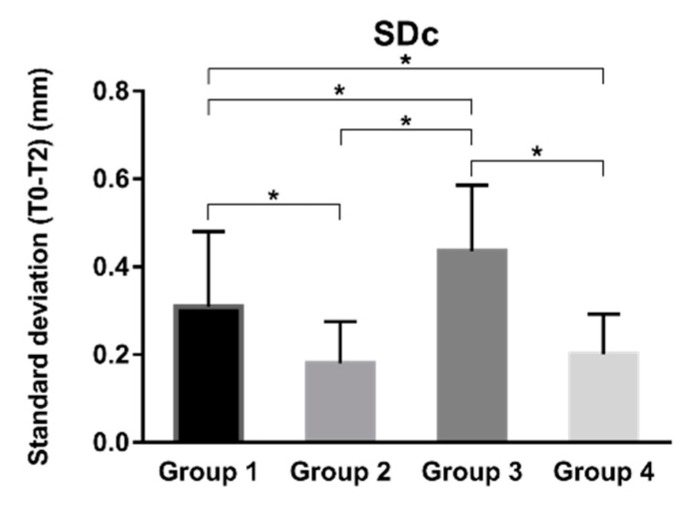
Results of the standard deviation of T0–T2 for different treatment modalities. Error lines represent +/− standard deviation. * Statistically significant.

**Table 1 materials-14-00583-t001:** Descriptive statistics for each group.

	Granulate (Group 1)	Granulate + Healing Cap (Group 2)	Sticky Bone (Group 3)	Customized Sticky Bone (Group 4)	*p* Value
**Patient demographics**					
Male/Female	5/7	7/5	7/5	6/6	0.919
Mean age ± SD	35.8 ± 13.9	30.9 ± 7.13	40.2 ± 13.7	35.4 ± 13.9	0.361
**Information about augmented sites**					
Number	13	12	19	19	
Location CI/LI/Canine	11/2/0	12/0/0	10/8/1	9/9/1	0.010 *
Jaw Maxilla/Mandible	13/0	11/1	17/2	15/4	0.342
Three-wall defect/Two-wall defect	9/4	12/0	14/5	11/8	0.062
Implant					0.000 *
NobelActive	10	4	11	5	
Bone Level Titanium SLA	3	7	1	2	
Tapered Screw-Vent MTX	0	0	0	1	
xiom REG	0	1	2	1	
Staged implant placement	0	0	5	10	
Bone substitutes					0.000 *
Bio-Oss	13	12	5	6	
Bio-Gene	0	0	14	13	

Abbreviations: SD, standard deviation; CI, central incisor; LI, lateral incisor. * Statistically significant.

**Table 2 materials-14-00583-t002:** Results of multi-factor ANOVA (*p* value).

Factor	Parameter						
	T0	T1	T2	T3	T4	T5	SD_C_
GBR procedures	0.001 *	0.001 *	0.003 *	0.001 *	0.000 *	0.000 *	0.000 *
Location	0.640	0.622	0.454	0.435	0.235	0.222	0.683
Jaw	0.289	0.433	0.449	0.627	0.677	0.723	0.074
Defect type	0.765	0.648	0.414	0.104	0.090	0.124	0.069
Implant type	0.888	0.816	0.684	0.514	0.142	0.091	0.588
Bone substitutes	0.213	0.367	0.364	0.461	0.214	0.038 *	0.051

Abbreviations: Tx, the thickness of labial bone graft measured x mm apical to the implant shoulder; SDc, the standard deviation of T0, T1 and T2. * Statistically significant.

**Table 3 materials-14-00583-t003:** Results of labial graft thickness after wound closure.

	Granulate (Group 1)	Granulate + Healing Cap (Group 2)	Sticky Bone (Group 3)	Customized Sticky Bone (Group 4)	Comparison between Groups
	Mean ± SD [95% CI]				
T0	2.77 ± 1.00	2.84 ± 0.74	2.95 ± 1.10	4.29 ± 0.92	1 vs. 2
	[2.16; 3.37]	[2.37; 3.31]	[2.42; 3.48]	[3.85; 4.73]	1 vs. 3
					1 vs. 4 *
					2 vs. 3
					2 vs. 4 *
					3 vs. 4 *
T1	3.01 ± 0.88	3.02 ± 0.75	3.59 ± 0.99	4.54 ± 0.88	1 vs. 2
	[2.48; 3.54]	[2.54; 3.50]	[3.11; 4.06]	[4.11; 4.96]	1 vs. 3
					1 vs. 4 *
					2 vs. 3
					2 vs. 4 *
					3 vs. 4 *
T2	3.39 ± 1.06	3.23 ± 0.75	4.00 ± 1.00	4.68 ± 0.79	1 vs. 2
	[2.75; 4.03]	[2.75; 3.71]	[3.52; 4.48]	[4.30; 5.06]	1 vs. 3
					1 vs. 4 *
					2 vs. 3
					2 vs. 4 *
					3 vs. 4 *
T3	3.41 ± 1.33	3.34 ± 0.73	4.26 ± 1.08	4.85 ± 0.82	1 vs. 2
	[2.61; 4.22]	[2.88; 3.81]	[3.74; 4.78]	[4.45; 5.24]	1 vs. 3 *
					1 vs. 4 *
					2 vs. 3 *
					2 vs. 4 *
					3 vs. 4
T4	3.50 ± 1.27	3.35 ± 0.69	4.47 ± 1.08	4.76 ± 0.96	1 vs. 2
	[2.73; 4.26]	[2.91; 3.79]	[3.95; 4.99]	[4.29; 5.22]	1 vs. 3 *
					1 vs. 4 *
					2 vs. 3 *
					2 vs. 4 *
					3 vs. 4
T5	3.46 ± 1.21	3.19 ± 0.70	4.61 ± 1.18	4.72 ± 0.98	1 vs. 2
	[2.73; 4.20]	[2.75; 3.64]	[4.05; 5.18]	[4.25; 5.19]	1 vs. 3 *
					1 vs. 4 *
					2 vs. 3 *
					2 vs. 4 *
					3 vs. 4
SD_c_	0.31 ± 0.17	0.18 ± 0.09	0.44 ± 0.15	0.20 ± 0.09	1 vs. 2 *
	[0.21; 0.41]	[0.12; 0.24]	[0.36; 0.51]	[0.16; 0.25]	1 vs. 3 *
					1 vs. 4 *
					2 vs. 3 *
					2 vs. 4
					3 vs. 4 *

Abbreviations: SD, standard deviation; 95% CI, 95% confidence interval; Tx, the thickness of labial bone graft measured x mm apical to the implant shoulder; SDc, the standard deviation of T0, T1 and T2. * Statistically significant.

## Data Availability

The data presented in this study are available on request from the corresponding author. The data are not publicly available due to protecting participant confidentiality.

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
