# Peer review of "The Influence of Different Guided Bone Regeneration Procedures on the Contour of Bone Graft after Wound Closure: A Retrospective Cohort Study"

_materials, 2021, doi:10.3390/ma14030583_

Round 1

Reviewer 1 Report

Dearest Author,

nice work, nothing relevant to say, just very minor suggestion to add a few comments and references on alternative biomaterials (see for example the reviwe of 2019 by Haugen et al) and some works on similar bovine-derived xenografts.

all the best & stay safe

Reviewer 2 Report

In this study, authors present a retrospective analysis of four different guided bone regeneration procedures for lateral ridge augmentation using bone grafts after wound closure. The thickness of labial graft was digitally measured just after wound closure and compared between the groups. The authors showed that the use of "sticky bone" in combination with surgical template showed the best outcome in terms of thickness of labial graft.

The study and results are very interesting and show the benefits of subject-specific planned surgeries to improve the outcome of bone augmentation. As such, I think it would be beneficial to expand on the digital technology applications for GBR procedures in the introduction (Lines 61-63) and discussion. What has been done so far and how? How widely used it is in clinical practice? What are the limitations and which patients may better benefit from that?

How was the surgical template fabricated? Was it 3D printed? It would be good to incorporate details of the manufacturing process and material used. Could the authors better describe the use of the surgical template in figure 4? What are the functions of the blue and green pieces?

The description of the CBCT analysis is a bit confusing. What is a prosthetically ideal position? Was the investigator designing such position also a trained surgeon? The paragraph would benefit from referring to the different sub-figures in figure 5.

In lines 168-170 authors mentioned that the measurements were performed on a cross-sectional image perpendicular to the virtual implant, how was that cross-section chosen?. A description of the different color showed in figures 5d and 5f is needed in the caption.

Line 255: Please define RCTs.

All the thickness measurements were limited to 2D as a single cross-section was analysed. This should be acknowledge in the limitations. Do the authors observed a variation of the thickness across the entire volume (i.e. all cross-sections). Why were not the CBCT images analysed as volumes instead of single cross-section?

Reviewer 3 Report

The subject of the article is very interesting and related to the modern and practiced implantology. The article was written in a correct, logical  and comprehensive language, the English is understandable, and the results provide an advance in current knowledge. The data and analyses are presented appropriately. In my opinion the highest standards for presentation of the results are used and the conclusions are interesting for the readership of the Journal. The results of the conducted tests were presented correctly and clearly. Numerous analyzes of the assessed parameters were used in the research. The literature review is rich and closely related to the subject of the article. The tables in the article help to understand the complexity of research.

Nevertheless some changes are necessary.

Please do not use any abbreviation in the title of the article.

The inclusion and exclusion criteria for patients need some correction: add to the inclusion criteria – good general health and absence of periodontal disease and remove from discussion criteria : uncontrolled systemic diseases, presence of acute infection, uncontrolled periodontal disease. Additionally add the occurrence of the uncontrolled systemic disease.

The exclusion criterion applies only to patients who meet the inclusion criterion, but for some important and extraordinary reason cannot take part in the studies.

Please change the time for centrifugation for 5 minutes.
